# Short-Term Effects of Ambient Air Pollution on ST-Elevation Myocardial Infarction Events: Are There Potentially Susceptible Groups?

**DOI:** 10.3390/ijerph16193760

**Published:** 2019-10-07

**Authors:** Hsiu-Yung Pan, Shun-Man Cheung, Fu-Cheng Chen, Kuan-Han Wu, Shih-Yu Cheng, Po-Chun Chuang, Fu-Jen Cheng

**Affiliations:** 1Department of Emergency Medicine, Kaohsiung Chang Gung Memorial Hospital, Chang Gung University College of Medicine, Kaohsiung City 833, Taiwan; gettingfat720@gmail.com (H.-Y.P.); fuchang@cgmh.org.tw (F.-C.C.); hayatowu1120@gmail.com (K.-H.W.); ma6021@cgmh.org.tw (S.-Y.C.); bogy1102@cgmh.org.tw (P.-C.C.); 2Chang Gung University College of Medicine, Guishan District, Taoyuan City 333, Taiwan; 3Department of Physical Medicine and Rehabilitation, Taitung Mackay Memorial Hospital, Taitung 950, Taiwan; man_c_man@yahoo.com.hk

**Keywords:** particulate matter, air pollution, ST-segment elevation myocardial infarction, emergency department, preexisting morbidity

## Abstract

Background: Air pollution exposure is associated with greater risk for cardiovascular events. This study aims to examine the effects of increased exposure to short-term air pollutants on ST-segment elevation myocardial infarction (STEMI) and determine the susceptible groups. Methods: Data on particulate matter PM2.5 and PM10 and other air pollutants, measured at each of the 11 air-quality monitoring stations in Kaohsiung City, were collected between 2011 and 2016. The medical records of non-trauma adult (>17 years) patients who had visited the emergency department (ED) with a typical electrocardiogram change of STEMI were extracted. A time-stratified and case-crossover study design was used to examine the relationship between air pollutants and daily ED visits for STEMI. Results: An interquartile range increment in PM2.5 on lag 0 was associated with an increment of 25.5% (95% confidence interval, 2.6%–53.4%) in the risk of STEMI ED visits. Men and persons with ≥3 risk factors (male sex, age, hypertension, diabetes, current smoker, dyslipidemia, history of myocardial infarction, and high body mass index) for myocardial infarction (MI) were more sensitive to the hazardous effects of PM2.5 (interaction: *p* = 0.039 and *p* = 0.018, respectively). The associations between PM10, NO2, and O3 and STEMI did not achieve statistical significance. Conclusion: PM2.5 may play an important role in STEMI events on the day of exposure in Kaohsiung. Men and persons with ≥3 risk factors of MI are more susceptible to the adverse effects of PM2.5 on STEMI.

## 1. Introduction

Myocardial infarction (MI) is one of the major causes of mortality in the world. MIs usually result from coronary thrombosis, which is caused by a ruptured or eroded atherosclerotic plaque that results in a sudden and serious reduction in blood flow [1]. There are two types of MI, ST-segment elevation myocardial infarction (STEMI) and non-ST-segment elevation myocardial infarction (NSTEMI). STEMI is most often the result of plaque rupture with the complete blockage of a coronary artery, for which timely reperfusion therapy is recommended [2]. NSTEMI typically occurs due to coronary artery occlusion, as a consequence of excessive plaque burden with myocardial ischemia or the formation of thrombus and the rupture of plaques without complete coronary artery occlusion [3]. Despite improvements in medical treatment and the rate of reperfusion therapy, the in-hospital mortality of STEMI is about 7.2%–2.5% [4].

Recently, the hazardous effects of air pollution on cardiovascular and respiratory diseases, such as pneumonia, asthma, and heart attacks, have been demonstrated by many epidemiological studies [5,6,7]. Toxicological studies have also demonstrated that particulate matter (PM) exposure was associated with myocardial cytotoxicity and coagulation biomarkers disturbance [8,9]. Few investigations have concentrated on the association between air pollutions and MI, and the results are inconclusive [3,10,11]. Gardner et al. [3] have shown a positive association between fine PM (defined as PM with an aerodynamic diameter smaller than 2.5 µm; PM_2.5_) and STEMI, but not NSTEMI. However, Evans et al. [10] did not find the association among ozone (O_3_), PM_2.5_, and STEMI. Most of these studies used a case-crossover study design. For this type of study, the “time” of disease onset is very important. Compared with NSTEMI, STEMI is thought to be the more acute disease, because one of the diagnosis criteria of STEMI is new-onset electrocardiogram change, which changes as time goes by [12]. Furthermore, the health effects of air pollution seem to have seasonal and regional variations [13,14,15]. The inconsistencies in different regions and seasons are partly explained by community characteristics, such as background location [16], climate condition [1], and the ratio of elderly residents [17]. Previous studies have reported that patients with some comorbidities might have an increased risk of out-hospital cardiac arrest (OHCA) and pneumonia, considering the levels of air pollutants [5,18]. Limited studies have focused on the health effect of air pollution on STEMI, especially for patients with preexisting diseases.

Therefore, the present study has two specific objectives: To evaluate the effects of increased short-term exposure to PM_2.5_ and other air pollutions on the events of STEMI and to explore the potential triggering effects of PM_2.5_, especially in patients with preexisting diseases.

## 2. Methods

### 2.1. Study Population

This retrospective observational study was conducted between January 1, 2011, and December 31, 2016, in an urban tertiary medical center with an average of 72,000 emergency department (ED) visits per year. The medical records of non-trauma patients older than 20 years who visited the ED with a principal diagnosis of MI (International Classification of Diseases, Ninth Revision [ICD-9]: 410; ICD-10: I21.0-I21.3) were extracted from the ED’s administrative database. The medical records were reviewed by two trained emergency physicians (EPs). Here, the inclusion criteria were a diagnosis meeting the criteria of STEMI on the basis of American College of Cardiology/American Heart Association guidelines [19], diagnosis confirmed by a cardiologist, and emergency primary percutaneous coronary intervention (PCI) performed with clinical evidence of coronary artery occlusion. Demographic factors, such as sex, age, and risk factors for MI (including diabetes, hypertension, current smoker, dyslipidemia, history of MI, body mass index [BMI]), and symptom onset time (date and hour) were obtained from the medical records of the patients. This study was approved by our hospital’s institutional review board (201801507B0) and has been performed in accordance with the ethical guidelines of the 1964 Declaration of Helsinki and its later amendments or comparable ethical standards. For this type of study, informed consent from the subjects was not required.

### 2.2. Pollutant and Meteorological Data

The air pollutant monitoring data and meteorological data were acquired from the Taiwanese Environmental Protection Administration, which has been gathering the information from 11 air-quality monitoring stations in Kaohsiung City since 1994. The hourly concentrations of four “criteria” pollutants, including NO_2_ (by ultraviolet fluorescence), PM_10_ (by beta-ray absorption), PM_2.5_ (by beta-ray absorption), and O_3_ (by ultraviolet photometry) were obtained.

The daily average of air pollution from each monitoring station, the residence of STEMI patients from their medical records, and the computed 24-hour average levels of the pollutants from the nearest monitoring station were collected. Daily meteorological data, including average humidity and temperature from the monitoring stations, were also obtained.

### 2.3. Statistical Method

A time-stratified and case-crossover design, an alternative design of Poisson time series regression modelling, was used to analyze the data [20,21]. The day the STEMI symptom occurred was set as lag 0, the day before the episode was lag 1, and the day before lag 1 was lag 2, and so on. The impacts of environmental conditions on STEMI were investigated from lags 0 to 3 individually. The case-crossover study design was recommended [20,21] to study the effects of intermittent, transient exposures on the following risk of acute-onset events in close temporal proximity to exposure. Within-subject comparisons in the case-crossover design were performed between case and control periods, and the hazard ratio was estimated from exposures at the event time rather than control periods. Time stratification was made to select referent days as the days falling on the same day of the week, within the same month as the index day. The levels of air pollution during the case period were compared with the levels on all referent days. The method of time stratification was used to adjust the effects of long-term trends, seasonal variation, and day of the week [22]. Conditional logistic regression was used to estimate the odds ratios (ORs) and 95% confidence intervals (CIs) of the STEMI associated with PM_2.5_ mass and other air pollutants. Subgroup analyses were also performed to identify the most susceptible groups, with the parameters of age, sex, and underlying disease.

Exposure levels to air pollutants and meteorological variables, such as average daily temperature and humidity, were entered into the models as continuous variables. Each pollutant model was adjusted for average temperatures and humidity on the day of STEMI and during lag intervals. The ORs were calculated based on interquartile range (IQR) increments in PM_2.5_, PM_10_, NO_2_, and O_3_ exposure. Statistical significance was set at *p* < 0.05. All statistical analyses were performed with SPSS version 25.0 (IBM Corp, Armonk, NY, USA).

### 2.4. Unblinded Ethical Approval Statement

This study was approved by the institutional review board of Kaohsiung Chang Gung Memorial Hospital (201801507B0) and has been performed in accordance with the ethical guidelines of the 1964 Declaration of Helsinki and its later amendments or comparable ethical standards. For this type of study, informed consent from the subjects was not required.

## 3. Results

During the 6-year study period, a total of 1104 STEMI cases met the inclusion criteria. We excluded 203 cases because their addresses were not in Kaohsiung City. The remaining 898 patients comprised our study group. The demographic characteristics of the 898 patients are listed in Table 1.

Table 2 shows a summary of the daily mean concentrations of air pollutants and the meteorological factors in Kaohsiung during the study period. The average PM_2.5_ concentration over the study period was 41.2 ± 18.9 μg/m^3^. 

Figure 1 shows the year-round estimates of air pollutants on STEMI. An IQR increment in PM_2.5_ on lag 0 was associated with increments of 25.5% (95% CI, 2.6%–53.4%) in STEMI incidence. Meanwhile, an IQR increase in PM_10_, NO_2_, and O_3_ was not significantly associated with STEMI. The association of PM_2.5_ with OHCA was strongest on lag 0 but decreased gradually on later lags.

Figure 2 demonstrates the results of the stratified analysis used to elucidate the effects of PM2.5 on STEMI according to different seasons, demographic factors, and underlying diseases on lag 0. After adjusting for temperature and humidity, male sex (interaction *p* = 0.039) was found to be more sensitive to the harmful effects of PM2.5. Except for male sex, the ORs were higher in individuals with risk factors of MI, who are currently smoking (interaction *p* = 0.590) or with hypertension (interaction *p* = 0.326), diabetes (interaction *p* = 0.359), dyslipidemia (interaction *p* = 0.215), or a history of MI (interaction *p* = 0.116), but the differences did not achieve statistical significance. The ORs were lower for elder (≥60, interaction *p* = 0.352) and individuals with high BMI (≥25, interaction *p* = 0.701), but there were no statistically significant differences. Then, analysis for individuals with multiple risk factors of MI, elder, male sex, individuals who are currently smoking, or those with hypertension, diabetes, dyslipidemia, BMI ≥25, or a history of MI was performed. Compared with those with less than three risk factors, patients with more than three risk factors of MI (interaction *p* = 0.018) were found to be more sensitive to the harmful effect of PM_2.5_.

The exposure-response relationship between PM_2.5_ levels and the risk of STEMI was calculated. Figure 3 shows that elevated levels of PM_2.5_ are significantly associated with an increased risk of STEMI compared with lower levels (Q1, ≤24.0 μg/m^3^). Exposure to a Q3 level (PM_2.5_ 41.2 to53.9 μg/m^3^) was significantly associated with a 39.7% (95% CI, 4.4%–86.9%; *p* = 0.024) increase in the risk of STEMI. When PM_2.5_ achieved a Q4 level (>53.9 μg/m^3^), the risk for STEMI significantly elevated to 52.0% (95% CI, 11.8%–106.7%; *p* = 0.008), compared with lower levels.

## 4. Discussion

In this study, we estimated the effects of PM and other air pollutants on STEMI and found that PM_2.5_ may play an important role in STEMI events in Kaohsiung, Taiwan. Of all air pollutant exposures included in the analysis, the risk of STEMI following PM_2.5_ exposure was greater in men and in patients with more than three MI risk factors.

Recently, several epidemiological studies have revealed that PM is associated with STEMI [3,23,24]. Argacha et al. [23] investigated 11,428 STEMI cases and found that an elevation in PM_2.5_ level by 10 μg/m^3^ on lag 0 was associated with an increase in STEMI risk by 2.8% (95% CI, 0.3%–5.4%). Meanwhile, other studies did not find a positive association between PM and MI [10,25,26]. One possible reason might be that the PM components were different between different regions and the difference in PM components led to different impacts on health [15,27]. In the study of Altemose et al. [27], blood coagulation biomarkers, particularly von Willebrand factor and soluble platelet selectin, were most consistently associated with oil combustion. Meanwhile, the systemic oxidative stress biomarker 8-hydroxy-2′-deoxyguanosine was most frequently associated with vehicle and industrial combustion. In the study of Ueda et al. [15], chloride, elemental carbon, and organic carbon were positively associated with cardiovascular mortality, and sulfate nitrate, in addition to ammonium, were correlated with respiratory mortality. The other possible reason might be that the difference in climate and temperature between different regions and the synergy of climate and temperature may lead to different hazards. Hsu et al. [28] showed that PM_2.5_ was associated with cardiovascular hospitalization, with stronger effects observed in the winter and on low-temperature days. Huang et al. [29] also demonstrated that the combinations of high levels of PM with low temperatures (<21 °C) were also associated with an increased incidence of acute coronary syndrome. The interactions among temperature, climate, and air pollution might lead to different study results in different regions. The study design might be another reason. There are two epidemiolocal methods that are commonly used to evaluate the short-term impact of pollution on health. Case-crossover designed was described by Maclure [30] and posed the advantage of controlling for potential confounding caused by time-invariant individual characteristics (such as age, sex, body mass index and comorbidities.) The major drawback of this method is that it is sensitive to the selected control period. [20,21] Most studies have used the case-crossover design to evaluate the impact of air pollution on health. In this design, the timing of disease diagnosis and disease onset are particularly important.

Owing to this reason, we used a time-series design based on a Poisson regression in our study. Our study’s inclusion criteria were rigorous, and we reviewed every medical chart manually to confirm the onset time. We found that PM_2.5_ may play an important role in STEMI events. 

Animal studies have shown that acute exposure to particulate air pollutants might lead to the activation of platelets and coagulation enzymes. It has been shown that ultrafine polystyrene particle exposure by intratracheal instillation in hamsters may enhance platelet activation and consequent thrombus formation [31]. The pulmonary deposition of PM plays an important role in that rapid activation of circulating platelets [32]. A significant increase in the number of circulating platelets was found in mice that were intratracheally exposed with diesel exhaust particles, and prothrombotic events were significantly aggravated in hypertensive mice also exposed to diesel exhaust particles [33]. Marchini et al. found that PM worsened the course of MI in mice by enhancing leukocyte recruitment, activating myeloid and endothelial cells, and inducing the production of the proinflammatory cytokines [34].

In humans, several hemostasis components in plasma were increased after increased exposure to air pollutions. For instance, Hassanvand et al. [8] have reported the increased production of proinflammatory cytokines and coagulation biomarkers after PM exposure. PM exposure was also found to be associated with endothelial dysfunction, platelet aggregation, and prothrombotic and antifibrinolytic effects, which might trigger thrombosis and MI [35]. It has been shown that an increase in coagulation and platelet activation, in addition to decreased fibrinolysis, results in venous and arterial thrombosis [36,37,38]. The effect was prominent for ST-elevation infarction, since STEMI typically involves plaque rupture and thrombus formation, which results in coronary artery occlusion [39]. Our study also supported the result that PM_2.5_ may play an important role in STEMI events.

Studies focused on the subgroup analysis of the effects of PM on STEMI are limited. Gardner et al. [3] demonstrated that patients with preexisting hypertension and other MI risk factors, such as dyslipidemia and prior MI, appeared to be more susceptible to the hazardous effects of PM_2.5_; however, statistical significance was only observed in the hypertension subgroup. Similar results were observed in our study. We found that patients with the risk factors of MI, such as hypertension, diabetes, dyslipidemia, current smoker, high BMI (≥25), or a previous history of MI, were more susceptible to the hazardous effects of PM_2.5_ for STEMI, but only the difference in sex was statistically significant (interaction *p* = 0.039). Men were reported to be more susceptible to PM_2.5_ for STEMI in one study, however, the interaction *p* value was not calculated in this instance [23]. Furthermore, in this study, we found that patients with ≥3 risk factors for MI were more susceptible to the hazardous effects of PM_2.5_ for STEMI than those with ≤2 risk factors (interaction *p* = 0.018). To the best of our knowledge, our study is the first to document that patients with multiple risk factors for MI are more susceptible to the hazardous effects of PM_2.5_.

The hazardous effect of PM seems to differ in different age groups, but the results are still inconclusive. Argacha et al. [23] have reported that patients of advanced age (≥75 years) are more susceptible to the hazardous effects of PM_10_ on the risk for STEMI, but they did not calculate the interaction *p* value. Gardner et al. [3] also noted the same susceptibility to the hazardous effects of PM_2.5_ on the risk for STEMI in patients with advanced age (≥65 years), but it was not statistically significant. Kang et al. [40] reported that patients with advance aged (≥60 years) were more susceptible to the hazardous effects of PM_2.5_ on the risk for OHCA. However, Collart et al. [41] found that the hazardous effect of PM_10_ on the risk for MI was more prominent in the younger group (aged 25–54 years). In addition, the lag pattern was different among different age groups, with a more delayed effect shown in the younger groups (25–54 years) when compared with the other two older groups (55–64 and ≥65 years). In our study, we found that younger patients (<60 years) were more susceptible to the hazardous effects of PM_2.5_ on STEMI, but the difference was not statistically significant. One possible reason for this difference is that PM exposure would increase the levels of antiangiogenic and proinflammatory cytokines [42], and the response might be different among different age groups [8]. For example, tumor necrosis factor soluble receptor II (sTNF-RII) was reported to play an important role in adverse health effect [43]. Hassanvand et al. [8] found that PM was not associated with sTNF-RII in healthy young adults, instead, that sTNF-RII was significantly associated with PM_2.5_ exposure in the elderly group [8]. Meanwhile, lifestyles might differ between the older and younger groups, hence, the amount of PM exposure would be varied in different age groups.

In many previous epidemiological studies, the time window for the health effects of air pollution ranged from 0 to 5 days. Gardner et al. [3] found that the risk of STEMI was significantly associated with an increase in PM_2.5_ concentration in the previous hour. The study of Liu et al. [11] found that increase in PM_2.5_ concentration at lags 2, 3, 4, and 0–5 days corresponded with increases in STEMI admissions. Another study found that the hazardous effect of PM_10_ on acute MI hospitalization was on lags 2 and 3 [41]. In our study, we found that PM_2.5_ exposure corresponds to ED visits for STEMI on lag 0. The difference could be caused by the different patient groups we enrolled. The studies of Collart et al. [41] and Liu et al. [11] enrolled patients admitted because of MI. In our study, we studied patients with ED visits due to STEMI. A time gap might exist between ED visit and admission, which might result in the difference in lag time. Second, the difference in the composition of PM between different regions might result in difference in time lag. Short-term exposure to PM_2.5_, PM_1–2.5_, and PM_1_ was associated with coagulation and inflammation blood markers, however, the associations were dependent on the PM size and differed across the various time lags [8]. The discrepancies in inflammation and coagulation blood markers that resulted from different PM sizes between different regions might lead to discrepancies in time lag. Third, the difference in the composition of population may result in different responses following PM exposure. Hassanvand et al. [8] reported differences in responses after PM exposure among different age groups [8]. For example, an increased level of interleukin 6 was observed at lag 1 in healthy young adults following PM_2.5_ exposure, however, interleukin 6 elevation was observed at lags 0, 1, 2, 3, and 4 in the older (aged >65 years) group. The elevated level of inflammation and coagulation biomarkers might correlate with the risk of STEMI. Therefore, the discrepancies of demographic structures among the different regions studied might result in differences in the responses following PM exposure, leading to different results.

The adverse health effect of PM appears to vary by seasons. Bell et al. [13] collected data from 202 US cities and found that the association of respiratory admission and cardiovascular admission with PM_2.5_ was strongest in the winter. Another study described an association between low temperature (<21 °C) in combination with high PM and occurrence of acute coronary syndrome [29]. Ueda et al. [15] demonstrated that PM_2.5_ was correlated with mortality, especially during transitional seasons (spring and autumn) [15]. A positive association between PM_2.5_ and pneumonia with septicemia during warm seasons was reported [5]. Evans et al. [10] reported that the association of PM_2.5_ with STEMI was stronger in the warm seasons (May–October). The findings were inconsistent in different studies. The discrepancy might be caused by different pollutant compositions in different seasons or different sources of pollutants in different cities [16,44]. In our study, a higher OR was found during the warm season, but the difference did not reach statistical significance (interaction *p* = 0.30).

The standard of PM_2.5_ was not consistent between different countries. The standard proposed by the Environmental Protection Agency (EPA) is 12 μg/m^3^ [45]. When PM_2.5_ was defined between 10 and 15 µg/m^3^, a previous study found significantly increased risk for OHCA, in comparison with the risk when the value set was set as <10 μg/m^3^ [39]. We found that the risk of STEMI was higher when the concentrations of PM were higher, namely, 39.7% higher in Q3 (41.2–53.9 μg/m^3^) and 52% higher in Q4 (>53.9 μg/m^3^), compared with Q1 (<24.0 μg/m^3^). The impact of PM on the risk of STEMI seemed to be dose-dependent, therefore, one should aggressively reduce the concentration of PM_2.5_.

### Study Limitations

Our study has some limitations. It was done in one industrial and tropical metropolitan city. Our findings might not be generalized to other locations that are meteorologically and ethnically different. Furthermore, factors such as the time spent outdoors, and the use of personal protective equipment and air purification may influence one’s exposure to pollutants and the magnitude of the observed associations when comparing with those at other geographical locations. Future studies should be performed in more regions with the use of personal protective equipment to overcome the limitations.

## 5. Conclusions

PM_2.5_ may play an important role in STEMI events on the day of exposure in Kaohsiung. Men and persons with ≥3 risk factors of MI were more susceptible to the adverse effects of PM_2.5_ on STEMI. Besides, we hope that the results of the study would motivate the government to pay closer attention to managing air pollution, especially PM_2.5_-based pollution.

## Figures and Tables

**Figure 1 ijerph-16-03760-f001:**
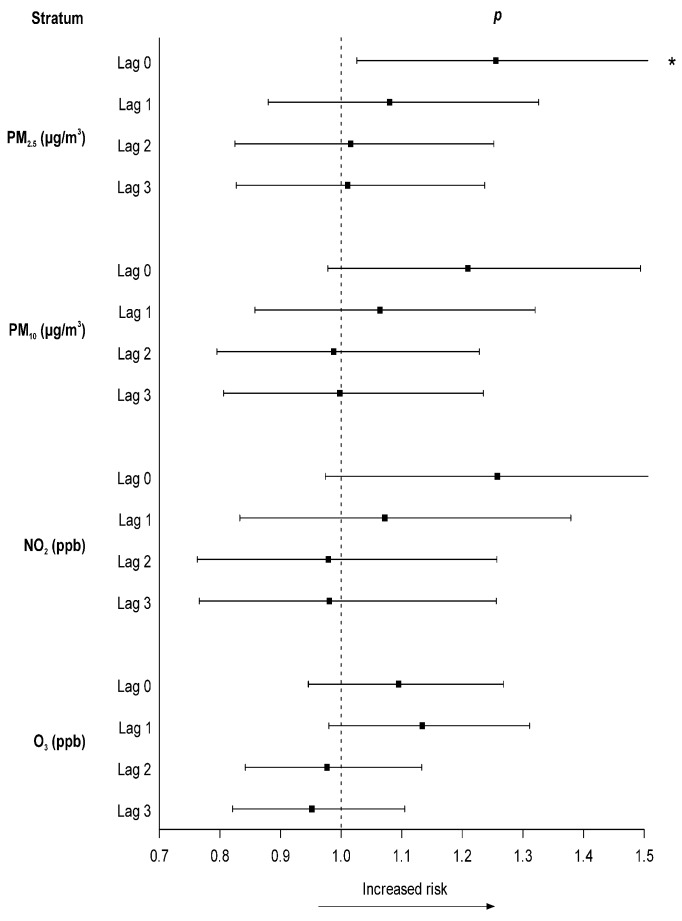
Odds ratio (95% confidence interval) for emergency department visits with ST-segment elevation myocardial infarction, associated with the interquartile range increase in air pollutants. Adjustments were made for temperature and humidity. PM: Particulate matter.

**Figure 2 ijerph-16-03760-f002:**
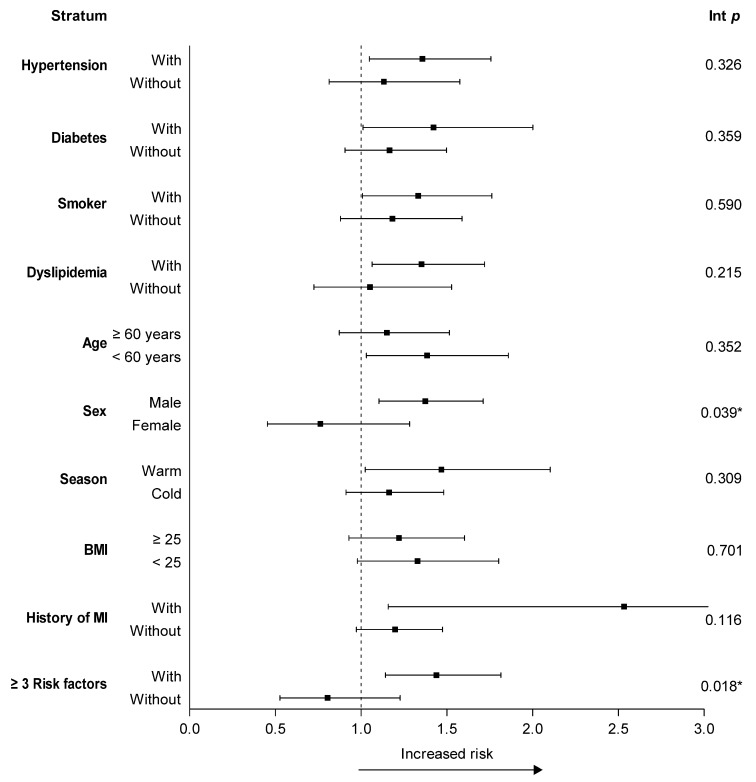
Odds ratios after adjusting for temperature and humidity, with an interquartile increase in PM_2.5_. * *p* < 0.05. Int *p*, interaction *p*-value. PM: Particulate matter. BMI: Body mass index. MI: Myocardial infarction.

**Figure 3 ijerph-16-03760-f003:**
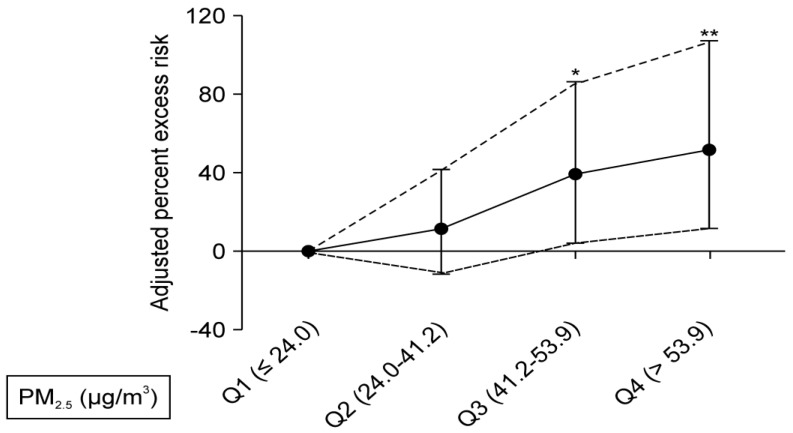
Adjusted risk of ST-segment elevation myocardial infarction according to ambient PM_2.5_ levels. The *y*-axis represents the percentage of excess risk with 95% confidence intervals. * *p*< 0.05, ** *p* < 0.01.

**Table 1 ijerph-16-03760-t001:** Demographic characteristics of patients (*n* = 898).

Characteristics	Number (%)
Age (years)	60.3 ± 12.8
Body mass index	25.6 ± 3.9
Male sex	745 (83.0)
Hypertension	572 (63.7)
Diabetes	338 (37.6)
Current smoker	491 (54.7)
Dyslipidemia	651 (72.5)
History of myocardial infarction	62 (6.9)
Risk factors ≧3	706 (78.6)

Data are presented as mean ± standard deviation or *n* (%).

**Table 2 ijerph-16-03760-t002:** Summary statistics for meteorological factors and air pollution in Kaohsiung, 2011–2016. PM: Particulate matter.

Air Pollutants and Meteorological Factors	Minimum	Percentiles			Maximum	Mean ± SD
		25%	50%	75%		
PM_2.5_ (µg/m^3^)	11.10	24.00	41.20	53.90	126.70	41.2 ± 18.9
PM_10_ (µg/m^3^)	19.60	43.30	69.70	93.30	190.00	70.9 ± 30.8
NO_2_ (ppb)	5.90	13.20	18.10	23.30	39.30	18.6 ± 6.5
O_3_ (ppb)	3.50	18.10	26.90	37.90	74.60	28.6 ± 12.8
Temperature (°C)	12.40	22.20	26.30	28.70	32.10	25.1 ± 4.3
Humidity (%)	47.50	70.90	74.80	78.90	94.40	75.1 ± 6.7

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
