# Peer review of "Short-Term Effects of Ambient Air Pollution on ST-Elevation Myocardial Infarction Events: Are There Potentially Susceptible Groups?"

_ijerph, 2019, doi:10.3390/ijerph16193760_

Round 1

Reviewer 1 Report

Although small in size (< 1000) and environment (one city) this study is able to discriminate PM2.5 as descriptor of STEMI, and even shows a dose response curve (Fig 3) using different quartiles of exposure around a high average level of 41 ug/m3 for PM2.5. Since such concentrations are quite exceptional over a longer period this merits some further discussion.

In addition I feel the paper should elaborate a bit more on the >3 risk factors. Please  explain the parameters in the abstract and also explain the reader why this cut-off is chosen, and whether this really arbitrary. Meaning is there no risk factor that scores higher than others. Fig 2 clearly suggests that previous MI is a higher risk than other risk factors.

Author Response

Point 1

Although small in size (< 1000) and environment (one city) this study is able to discriminate PM2.5 as descriptor of STEMI, and even shows a dose response curve (Fig 3) using different quartiles of exposure around a high average level of 41 ug/m3 for PM2.5. Since such concentrations are quite exceptional over a longer period this merits some further discussion.

Response 1 : Thank you for pointing this out.

We added a part of discussion in line 283 to line 289, as following:

“The standard of PM2.5 was not consistent between different countries. The standard proposed by the Environmental Protection Agency (EPA) is 12 μg/m3 [44]. When PM2.5 was defined between 10 and 15 µg/m3, a previous study found significantly increased risk for OHCA, in comparison with the risk when the value set was set as <10 μg/m3 [39]. We found that the risk of STEMI was higher when the concentrations of PM were higher: 39.7% higher in Q3 (41.2–53.9 μg/m3) and 52% higher in Q4 (>53.9 μg/m3), compared with Q1 (<24.0 μg/m3). The impact of PM on the risk of STEMI seemed to be dose-dependent; therefore, one should reduce the concentration of PM2.5 aggressively.”

Point 2

In addition I feel the paper should elaborate a bit more on the >3 risk factors. Please  explain the parameters in the abstract and also explain the reader why this cut-off is chosen, and whether this really arbitrary. Meaning is there no risk factor that scores higher than others. Fig 2 clearly suggests that previous MI is a higher risk than other risk factors.

Response 2:

Thank you for the comment.

We added explanations in part of abstract, line 29-30, as following:“An interquartile range increment in PM2.5 on lag 0 was associated with increments of 25.5% (95% confidence interval, 2.6%–53.4%) in the risk of STEMI ED visits. Men and persons with ³3 risk factors (male sex, age, hypertension, diabetes, current smoker, dyslipidemia, history of myocardial infarction, and high body mass index) for MI were more sensitive to the hazardous effects of PM2.5 (interaction: p=0.039 and p=0.018, respectively).”

We also explained why patients with multiple risk factors for MI was analyzed in line 152-164, as following:

“Figure 2 demonstrates the results of the stratified analysis used to elucidate the effects of PM2.5 on STEMI according to different seasons, demographic factors, and underlying diseases on lag 0. After adjusting for temperature and humidity, male sex (interaction p=0.039) was found to be more sensitive to the harmful effects of PM2.5. Except for male sex, the ORs were higher in individuals with risk factors of MI, who are currently smoking (interaction p=0.590) or with hypertension (interaction p=0.326), diabetes (interaction p=0.359), dyslipidemia (interaction p=0.215), or a history of MI (interaction p=0.116), but the differences did not achieve statistical significance. The ORs were lower for elder (≥60, interaction p=0.352) and individuals with high BMI (≥25, interaction p=0.701), but there were no statistically significant differences. Then, analysis for individuals with multiple risk factors of MI, elder, male sex, individuals who are currently smoking, or with hypertension, diabetes, dyslipidemia, BMI≥25, or a history of MI was performed. Compared with those with less than three risk factors, patients with ≥3 risk factors of MI (interaction p=0.018) were found to be more sensitive to the harmful effect of PM2.5.”

Reviewer 2 Report

I'm sorry but I think that your interesting work must be submitted to mayor revision related to the statistical methods. I think that, because the relevant number of observed STEMI is necessary to use a "multivariate" statistical approach and not a "stratified" one.

The results showed in the Fig.2 are, in fact, clear: the associations (ORs) between PM 2.5 and STEMI are not significant for the stratum without all the risk factors (Hypertension, Diabetes, Smokers, Dyslipidemia, Age, Sex, Season, BMI, History of MI). 

It is possible that, using a multivariate unconditional logistic regression method, the results could be modified toward the conclusions speaking about "susceptibility" (?) of the adverse effects of PM 2.5 on STEMI.

Good work!

Author Response

Point 1:

I'm sorry but I think that your interesting work must be submitted to mayor revision related to the statistical methods. I think that, because the relevant number of observed STEMI is necessary to use a "multivariate" statistical approach and not a "stratified" one.

Response 1 :  Thank you for pointing this out.

The aim of the study was to investigate the relationship between air pollution and ED visit for STEMI. For this kind of study, case-crossover study design was widely accepted. The case-crossover study design was recommended [20,21] to study the effects of intermittent, transient exposures on the following risk of acute-onset events in close temporal proximity to exposure. This design is an adaptation of the case-control study in which each case serves as his or her own referent. In the case-crossover design, only cases are sampled and the hazard ratio is estimated from within-subject comparisons of exposures at the event time than control periods of the event. As the result, “multivariate” statistical approach was not suitable for the present study. We also added the following explanation in the part of statistical method, line 104-108:

“The case-crossover study design was recommended [20,21] to study the effects of intermittent, transient exposures on the following risk of acute-onset events in close temporal proximity to exposure. Within-subject comparisons in the case-crossover design were performed between case and control periods, and the hazard ratio is estimated from exposures at the event time than control periods.

Point 2:

The results showed in the Fig.2 are, in fact, clear: the associations (ORs) between PM 2.5 and STEMI are not significant for the stratum without all the risk factors (Hypertension, Diabetes, Smokers, Dyslipidemia, Age, Sex, Season, BMI, History of MI). 

Response 2:  Thank you for the comment.

  In figure 1, we found that exposure to PM 2.5 would result in increased risk of STEMI, we conducted a subgroup analysis then. After adjusting for temperature and humidity, male sex (interaction p=0.039) was found to be more sensitive to the harmful effects of PM2.5. For patients without hypertension, DM…etc., the ORs had not reached statistically significance, however, most of ORs value was >1. It meant that the impact of PM2.5 exposure was differed between different subgroups but the hazard effect of PM2.5 was greater on some subgroups (male, risk factor≥3). We

“Figure 2 demonstrates the results of the stratified analysis used to elucidate the effects of PM2.5 on STEMI according to different seasons, demographic factors, and underlying diseases on lag 0. After adjusting for temperature and humidity, male sex (interaction p=0.039) was found to be more sensitive to the harmful effects of PM2.5. Except for male sex, the ORs were higher in individuals with risk factors of MI, who are currently smoking (interaction p=0.590) or with hypertension (interaction p=0.326), diabetes (interaction p=0.359), dyslipidemia (interaction p=0.215), or a history of MI (interaction p=0.116), but the differences did not achieve statistical significance. The ORs were lower for elder (≥60, interaction p=0.352) and individuals with high BMI (≥25, interaction p=0.701), but there were no statistically significant differences. Then, analysis for individuals with multiple risk factors of MI, elder, male sex, individuals who are currently smoking, or with hypertension, diabetes, dyslipidemia, BMI≥25, or a history of MI was performed. Compared with those with less than three risk factors, patients with ≥3 risk factors of MI (interaction p=0.018) were found to be more sensitive to the harmful effect of PM2.5.”

Point 3:

It is possible that, using a multivariate unconditional logistic regression method, the results could be modified toward the conclusions speaking about "susceptibility" (?) of the adverse effects of PM 2.5 on STEMI.

Response 3: Thank you for the comment.

  The case-crossover study design was used in the present study. The inference is based on a comparison of each subject's exposure during a time period relevant for the causation of the outcome, often referred to as a hazard period, and during one or more control periods. The analytic approach is analogous to crossover and matched case-control studies. The hazard ratio is estimated from within-subject comparisons of exposures at the event time than control periods. Conditional logistic regression analysis was then used to estimate adjusted odds ratios. For this study design, multivariate unconditional logistic regression could not be applied. On the other hand, case-crossover design is amenable for studying the effects of varying short-term air pollution exposure on health outcomes (Jaakkola, 2003), and case-crossover design was widely accepted in many studies of air pollution on health effects (Darrow et al., 2014; Evans et al., 2017; Pope et al., 2006).

Previous studies focused on the subgroup analysis of the effects of PM on STEMI are limited. In our study, we found that patients with the risk factors of MI, such as hypertension, diabetes, dyslipidemia, current smoker, high BMI (≥25), or previous history of MI, were more susceptible to the hazardous effects of PM2.5 for STEMI, but only the difference in sex was statistically significant (interaction p=0.039). In our study, patients with ³3 risk factors for MI were found to be more susceptible to the hazardous effects of PM2.5 for STEMI than those with ≤2 risk factors (interaction p=0.018). To the best of our knowledge, our study is the first to document that patients with multiple risk factors for MI were more susceptible to the hazardous effects of PM2.5. We observed that younger patients (<60 years) were more susceptible to the hazardous effects of PM2.5 on STEMI, but the difference was not statistically significant. One possible reason for the difference was that PM exposure would increase the levels of antiangiogenic and proinflammatory cytokines and the response might be different among different age groups.

Darrow, L. A., Klein, M., Flanders, W. D., Mulholland, J. A., Tolbert, P. E., & Strickland, M. J. (2014). Air pollution and acute respiratory infections among children 0-4 years of age: an 18-year time-series study. Am J Epidemiol, 180(10), 968-977. doi:10.1093/aje/kwu234

Evans, K. A., Hopke, P. K., Utell, M. J., Kane, C., Thurston, S. W., Ling, F. S., . . . Rich, D. Q. (2017). Triggering of ST-elevation myocardial infarction by ambient wood smoke and other particulate and gaseous pollutants. J Expo Sci Environ Epidemiol, 27(2), 198-206. doi:10.1038/jes.2016.15

Jaakkola, J. J. K. (2003). Case-crossover design in air pollution epidemiology. European Respiratory Journal, 21(Supplement 40), 81S-85s. doi:10.1183/09031936.03.00402703

Pope, C. A., 3rd, Muhlestein, J. B., May, H. T., Renlund, D. G., Anderson, J. L., & Horne, B. D. (2006). Ischemic heart disease events triggered by short-term exposure to fine particulate air pollution. Circulation, 114(23), 2443-2448. doi:10.1161/CIRCULATIONAHA.106.636977

Round 2

Reviewer 2 Report

I'm sorry but i don't agree with your answers  and the modified text is very similar to the original one.